# A handheld fiber-optic tissue sensing device for spine surgery

**Merle S. Losch**[1]*, **Benjamin E. Visser**[1], **Jenny Dankelman**[1], **Benno H. W. Hendriks**[1,2]

**1** Department of Biomechanical Engineering, Faculty of Mechanical Engineering, Delft University of Technology, Delft, The Netherlands, **2** Philips Medical Systems Nederland B.V., Best, The Netherlands

* m.s.losch@tudelft.nl

**Data Availability Statement:** The corresponding CAD files and PCB schematic for this publication can be found here: doi.org/10.4121/77363d00-f68c-4da7-b32e-3aabe957ef98.

## Abstract

The demographic shift has increased the demand for surgical interventions to address age-related degenerative diseases, such as spinal fusion. Accurate placement of pedicle screws, crucial for successful spinal fusion, varies widely with physician experience. Integrating tissue sensing into spine surgical instruments allows intraoperative examination of tissue properties, providing surgeons with additional information to prevent screw misplacement. This paper introduces a handheld fiber-optic tissue sensing device for real-time bone tissue differentiation during spine surgery using Diffuse Reflectance Spectroscopy (DRS). Our prototype employs laser diodes at two distinct wavelengths for tissue illumination, eliminating the need for a spectrometer and enabling direct light collection with a photodiode. The device includes a printed circuit board (PCB) with driver circuits that are adjustable for varying laser diode output power, and signal amplification to convert the photodiode current to a measurable voltage signal. Controlled by a microcontroller, the device computes a reflectance ratio from both laser diode signals to provide real-time audio feedback to surgeons across various healthcare settings. Despite challenges in coupling efficiencies from manual fiber-coupling of the diodes, our prototype is able to emit and collect light to distinguish bone tissues with DRS, demonstrating feasibility. It is compact, made of low-cost and readily available components, and offers fast, real-time feedback, thus serving as a successful proof-of-concept for enhancing surgical accuracy during spinal fusion procedures.

## Introduction

Over the past decades, the world has witnessed a demographic shift towards increased life expectancy, accompanied by a growing need for surgical interventions to address age-related degenerative diseases [1, 2]. Among these interventions, spinal fusion is an option for patients seeking relief from debilitating spinal conditions. Central to the success of spinal fusion procedures is the accurate placement of pedicle screws, which provide stability to the spine during the fusion process [3]. Despite the emphasis on achieving surgical accuracy to avoid complications and revision surgeries, accurate screw placement remains a challenge due to the complex anatomy of the spine and the reliance on tactile feedback and anatomical expertise [4, 5].

**Funding:** This work was supported by the grant: NWO-TTW 17553. The funders had no role in study design, data collection and analysis, decision to publish, or preparation of the manuscript.

**Competing interests:** The author affiliated with Philips Medical Systems (B.H.W.H.) has financial interests in the subject matter, materials, and equipment, in the sense that he is an employee of Philips. None of the other authors have any financial relationship or conflict of interests. This does not alter our adherence to PLOS ONE policies on sharing data and materials.

Misplacement of pedicle screws can lead to severe complications including neurological injury, and the need for revision surgery [6]. To enhance the accuracy of screw placement, various guidance systems such as intraoperative fluoroscopy, computer-assisted navigation, and robotic assistance have been developed [7]. While these technologies have shown great promise in improving accuracy, they come with drawbacks including high costs, the need for specialized training, and limitations in availability [8–10].

Intra-operative ultrasonography is cost-effective and provides real-time imaging within a portable device. However, ultrasound has limitations in visualizing bony structures due to poor penetration through dense tissue and low signal-to-noise ratio (SNR), and its efficacy still largely depends on the operator's expertise [11, 12]. These limitations underscore the need for alternative approaches that offer more direct, consistent, and interpretable feedback during spine surgery.

Integrating tissue sensing into surgical instruments presents a promising alternative for improving the accuracy of pedicle screw placement. By examining the physical properties of surrounding tissue in real-time, tissue sensing devices offer valuable feedback to surgeons, allowing them to adjust screw trajectories and prevent misplacement. While the PediGuard system (Spine-Guard SA, Vincennes, France) has pioneered tissue sensing in spine surgery using Electrical Impedance Spectroscopy (EIS), it faces challenges such as susceptibility to blood and tissue accumulation in the pilot hole [13, 14]. Additionally, as the device only measures tissue in direct contact with its tip, a cortical breach can only be identified once the breach is already in progress, limiting the opportunity for early intervention [13, 15]. Given these limitations, Diffuse Reflectance Spectroscopy (DRS) emerges as a compelling alternative.

DRS operates on the principle of light interaction with tissue to assess its composition. A spectrum of light is directed into tissue via optical fibers, inducing scattering and absorption processes that vary according to the tissue's structure and biochemical composition. The light that is reflected back carries information about the tissue's optical properties. Studies have demonstrated the effectiveness of DRS in distinguishing between cancellous and cortical bone based on fat content, thereby providing reliable feedback for pedicle screw placement [16, 17]. This technology offers real-time feedback and the potential for integration into existing surgical devices through fiber-optic technology [18–22]. DRS enables early breach detection and remains unaffected by blood interference when using near-infrared (NIR) wavelengths [23, 24]. However, current DRS systems suffer from cumbersome and expensive setups which at present limit their application in the operating room.

In this paper, we present a handheld fiber-optic tissue sensing device for spine surgery. Unlike traditional DRS systems, our device uses only two distinct wavelengths instead of broadband light for illumination. This eliminates the need for the spectrometer typically used in DRS systems to separate the spectrum of the collected light, as light can be collected directly by a simple photodetector, making our system compact and affordable. Controlled by a microcontroller, our device provides real-time audio feedback to surgeons. Through this novel design, we aim to democratize the availability of tissue sensing technology, making it accessible to a wider range of healthcare practitioners and facilities.

## System components and design

The design of the handheld fiber-optic tissue sensing device was driven by three main considerations:

1. **size**: The components must be compact to accommodate placement within a handheld device.

2. **cost**: The components must be readily available off-the-shelf or easily manufactured to ensure the device's affordability.

3. **speed**: The sensing device must provide real-time tissue feedback based on DRS measurements.

All components—a light source, a photodetector, the electronic circuits, a microcontroller, the feedback system, and the casing—were selected and designed with these three criteria in mind.

## Light source

DR spectra vary based on the measured tissue's structure and composition. To extract biologically relevant information, the collected spectrum can be compared against absorption spectra of key chromophores, such as collagen, lipids, and water. While DRS typically employs a broadband light source for tissue illumination, we opted for just two wavelengths. By specifically selecting wavelengths where cancellous and cortical bone tissues show significant differences in optical properties, we can effectively distinguish between them based on the ratio of their signals, avoiding wavelengths dominated by hemoglobin or water absorption, which could hinder tissue differentiation.

Previous studies have identified optimal wavelengths for bone tissue differentiation in DRS measurements [16, 18, 25–28]. For distinguishing cortical bone from other tissues, four wavelength regions within the visible/NIR spectrum were identified: 695-699 nm, 925-927 nm, 1188 nm, and 1207-1211 nm [29].

As light-emitting diodes (LEDs) typically have low power output, laser diodes were chosen as the light source due to their higher focused output power and narrow emission spectrum (5-20 nm). To meet size and cost constraints, we selected simple TO-can laser diodes over fiber-coupled ones, which are readily available off-the-shelf and offer a wide range of wavelength options in a typical compact 5.6 mm TO-can packaging. Considering cost-effectiveness and time efficiency, we avoided custom-ordering laser diodes and selected available wavelengths, including 690 nm, 905 nm, 940 nm, and 1270 nm. Additionally, 1310 nm was chosen as a reference wavelength with similar absorption in both cancellous and cortical bone. These wavelengths were input for Monte Carlo (MC) simulations in MCmatlab [30] to model light scattering and absorption within the bone tissue. The simulation results revealed the largest difference between cancellous and cortical bone in the intensities collected at 940 nm and 1310 nm, leading to their selection as the light sources for our device.

The chosen laser diodes, operating at 940 nm (QL94J6SA, Roithner Lasertechnik GmbH, Vienna, Austria) and 1310 nm (RLT1310-5MGS-P2, Roithner Lasertechnik GmbH, Vienna, Austria), Fig 1 ①, were selected based on their power ratings of 50 mW and 11 mW respectively. Both diodes provide sufficient power for a good SNR on tissue samples and are deemed safe for continuous illumination [31, 32].

To integrate DRS into a thin probe, we utilize 400 μm core diameter optical fibers with a numerical aperture (NA) of 0.22 (FG400LEA, Thorlabs Inc., Newton, NJ, USA). The laser diodes are coupled to the optical fibers using NIR-transparent optical adhesive (NOA81, Norland Products Inc., Jamesburg (NJ), USA).

**Driver circuit.** The two laser diodes are powered by electronic driver circuits that ensure constant output power. Automated power control (APC) circuitry enhances reliability and consistency, particularly in applications requiring precise power levels over extended durations, by stabilizing output and compensating for environmental variations. The TO-can packaging of the laser diodes used as the light source incorporates a photodiode to enable feedback

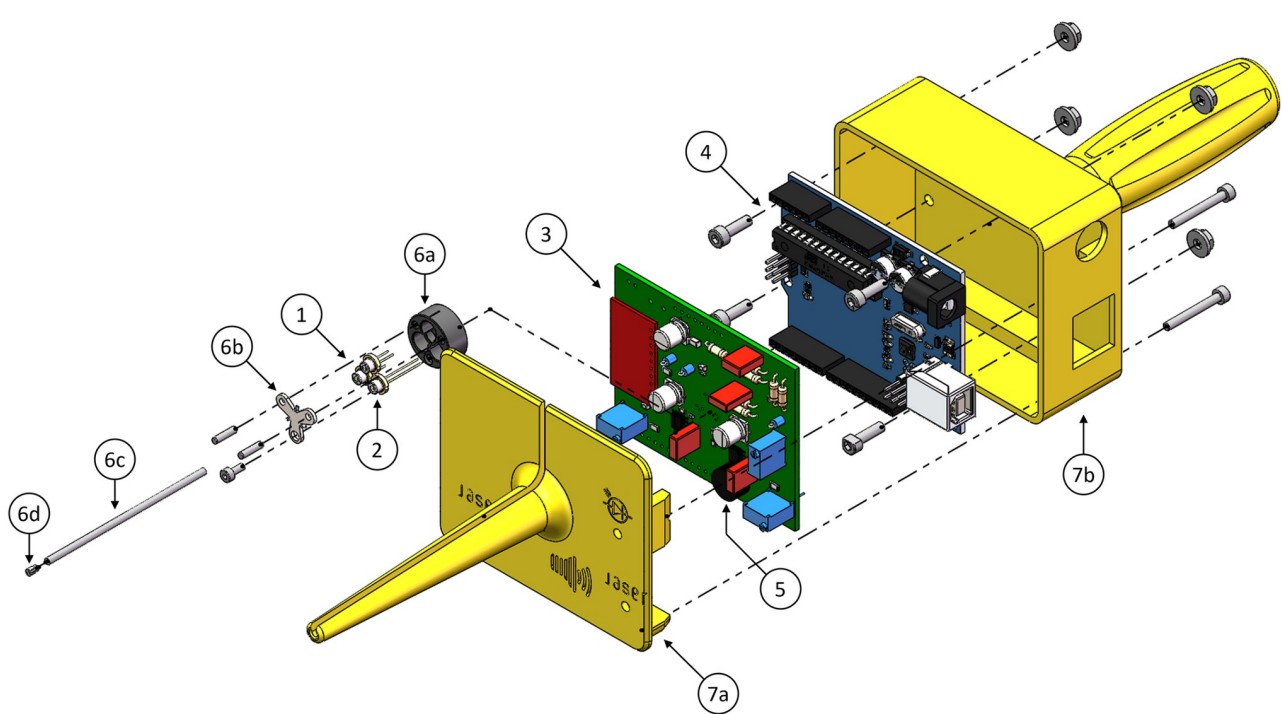

**Fig 1. Exploded view of handheld fiber-optic tissue sensing device including two laser diodes (1), one photodiode (2), a printed circuit board (PCB) (3), an Arduino (4), a buzzer (5), a diode holder unit (6a-d), and a two-part housing (7a+b).** The corresponding CAD files can be found here: doi.org/10.4121/77363d00-f68c-4da7-b32e-3aabe957ef98.

control of the laser's output power. This photodiode generates a photocurrent proportional to the laser diode's emission, facilitating the monitoring of laser power.

To achieve APC of the laser diodes, driver integrated circuits (ICs) (iC-WJ SO8, iC-Haus, Bodenheim, Germany) are employed. These ICs are designed for the N-type built-in configuration of the laser diodes and can operate in constant or pulsed mode up to 300 kHz. The IC sets the desired output through external resistors and capacitors. The schematic of the dynamic driver circuit is depicted in Fig 2.

The average optical output power, determined by the monitor current amplitude $I_{av}$(AMD), is set by adjusting $R_{SET}$ (potentiometer ranging up to 25 kΩ), with $R_{SET2}$ = 2.7 kΩ serving as the minimum value within this control range:

$$R_{SET} + R_{SET2} = \frac{CR \cdot V(\text{ISET})}{I_{av}(\text{AMD})}. \qquad [\Omega] \qquad (1)$$

The IC specifies a current ratio $CR$ of 1 and a constant voltage $V$(ISET) of 1.22V as its electrical characteristics. The expected monitor currents under continuous mode are 0.2 mA for the 940 nm laser diode and 0.3 mA for the 1310 nm laser diode. Operating the laser diodes with a pulse duty factor of 50% yields $R_{SET}$ values of 9.5 kΩ for 940 nm and 5.4 kΩ for 1310 nm.

To smooth out the control, a capacitor $C_I$ is employed. This prevents peak values from reaching the laser diode during pulse control, reducing the risk of damage. According to the IC's specifications, capacitor $C_I$ can be set to 100 nF in constant mode, while maintaining input pin IN at 5V. For pulse mode operation, input pin IN can be alternated between 0V and 5V at the desired frequency. As specified by the manufacturer, the capacitor's value should be increased with slower pulse repetition frequency $f$ or larger monitor current to accommodate

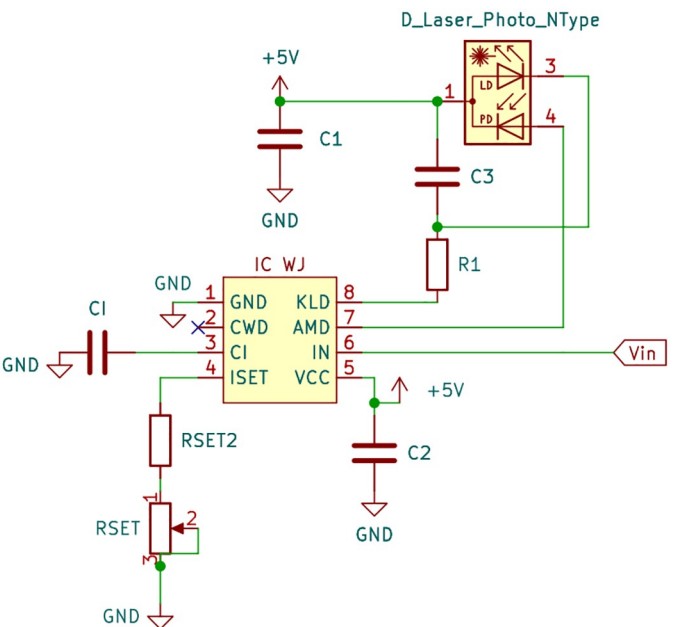

**Fig 2. Schematic of the dynamic driver circuit for N-type laser diodes.**

the larger charge buildup:

$$C_I \geq \frac{440 \cdot I(\text{ISET})}{f \cdot V(\text{ISET})} = \frac{440}{f \cdot (R_{SET} + R_{SET2})}. \qquad [\text{F}] \qquad (2)$$

Additionally, capacitors $C_1$ = 100 μF and $C_2$ = 0.1 μF serve to stabilize the power supply, preventing voltage fluctuations for increased reliability. Capacitor $C_3$ = 2 nF protects the laser diode from electrostatic discharge or transients that could cause damage. Finally, series resistor $R_1$ = 12 Ω reduces the total power consumption of the IC by limiting current and damping oscillations from the power supply.

## Photodetector

DRS relies on reflectance-based measurements, requiring that the light source and photodetector be placed side by side for in-vivo probing. The source-detector separation (SDS) is set at 1.3 mm to ensure sufficient look-ahead distance into the bone tissue while maintaining a compact setup and reasonable light attenuation in the tissue [16, 23].

The reflected light from the bone tissue can be directly measured by a photodiode. When illuminated by light, a photodiode generates an electric current. The output current $I_{PD}$ of the photodiode is proportional to the optical power $P_{in}$ incident on its surface. To ensure a linear relationship between optical power and output current, the photodiode operates in photoconductive mode (reverse-biased) rather than photovoltaic mode (zero bias). This choice increases the responsivity while possibly leading to a higher dark current due to the widened depletion junction. The photodiode current comprises both the signal current and the offset dark current.

The photodiode selected for our device is a wide spectral range InGaAs photodiode (IG17X500S4ix, Laser Components Germany GmbH, Olching, Germany), Fig 1 ②, which we

fiber-coupled in the same way as the aforementioned laser diodes. It exhibits varying responsivity $R_{PD}$ across the wavelength spectrum (500-1700 nm), with a peak responsivity of 1.05 A W$^{-1}$ observed around 1550 nm.

**Expected signal.** Using the wavelength-specific responsivity of the photodiode $R_{PD}(\lambda)$, the photodiode current is determined as follows:

$$I_{PD} = P_{in} \cdot R_{PD}(\lambda). \qquad [A] \qquad\qquad (3)$$

The incident optical power $P_{in}$ on the photodiode surface is calculated as:

$$P_{in} = P_{LD} \cdot \eta_{LD} \cdot R_{tissue}(\lambda) \cdot \eta_{PD}, \qquad [W] \qquad\qquad (4)$$

where $P_{LD}$ represents the emitted power of the laser diode, $\eta_{LD}$ denotes the laser-diode-to-light-guide coupling efficiency, $R_{tissue}(\lambda)$ refers to the percentage of emitted photons at a specific wavelength $\lambda$ that reach the collecting fiber after interaction with the tissue, and $\eta_{PD}$ denotes the light-guide-to-photodiode coupling efficiency. Table 1 gives an overview of the expected values for all parameters influencing the photodiode current.

The driver circuits were designed to emit light at a power of 51.1 mW for the 940 nm laser diode and 10.3 mW for the 1310 nm laser diode. Some of this light is lost due to butt-coupling, the process of bonding the diode to the fiber end with optical adhesive. Butt-coupling typically achieves a maximum efficiency of 50% [33–35]. We conservatively estimated the laser and photodiode coupling efficiencies at 30% each.

Since the incident optical power is dependent on tissue reflection, the fractions of collected photons (expressed as permille) for cancellous ($R_{canc}$) and cortical bone ($R_{cort}$) at 940 nm and 1310 nm were estimated through MC simulations of our setup. With emitter and collector simulated as optical fibers (core diameter 400 μm, NA = 0.22) at SDS = 1.3 mm, we simulated $10^9$ photons of a wavelength of 940 nm and 1310 nm each within either a single-layer cancellous bone model ($\mu_a(940nm) = 0.216$, $\mu'_s(940nm) = 28.6$, $\mu_a(1310nm) = 0.833$, $\mu'_s(1310nm) = 17.3$, $g = 0.9$ [16]) or a single-layer cortical bone model ($\mu_a(940nm) = 0.223$, $\mu'_s(940nm) = 16.4$, $\mu_a(1310nm) = 0.920$, $\mu'_s(1310nm) = 11.3$, $g = 0.9$ [16]) with the dimensions 3.0 mm x 3.0 mm x 3.0 mm and a resolution of 100 bins/mm (simulation run for 300 x 300 x 300 voxels) using MCMatlab [30]. The obtained reflectance values are ranging around 0.1‰-0.15‰ as listed in Table 1.

Photodiode responsivity is found in the datasheet as 0.62 A W$^{-1}$ at 940 nm and 0.91 A W$^{-1}$ at 1310 nm. Considering our estimation and the equations presented above, a projected photocurrent ranging from 81 nA to 432 nA is anticipated.

**Signal amplification.** Given the anticipated low levels of the photodiode current, a transimpedance amplifier (TIA) is necessary to amplify these currents into measurable voltage signals $V_{out}$ for further processing. The estimated maximum photodiode current plays a critical

**Table 1. Parameters affecting $I_{PD}$.**

|  | 940 nm | 1310 nm |
|---|---|---|
| $P_{LD}$ [mW] | 51.1 | 10.3 |
| $\eta_{LD}$ [%] | 30 | 30 |
| $\eta_{PD}$ [%] | 30 | 30 |
| $R_{canc}$ [‰] | 0.152 | 0.120 |
| $R_{cort}$ [‰] | 0.149 | 0.0955 |
| $R_{PD}$ [A W$^{-1}$] | 0.62 | 0.91 |

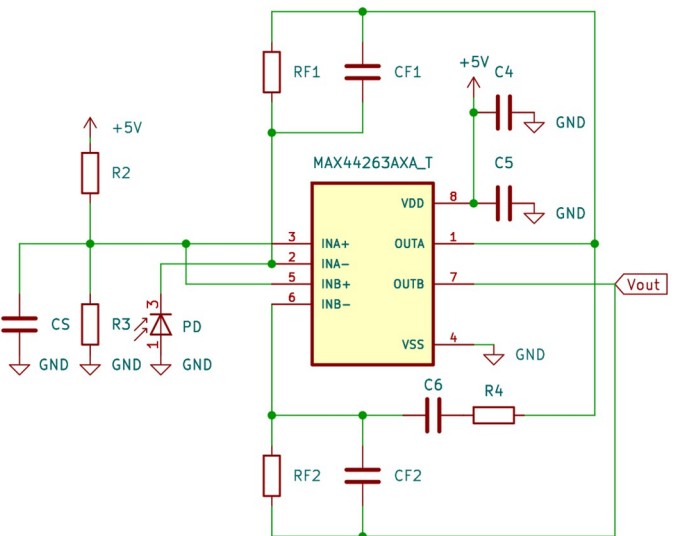

**Fig 3. Schematic of the two-stage, low-noise transimpedance amplifier circuit.**

role in determining the gain $G$ of the TIA:

$$V_{out} = I_{PD} \cdot G. \qquad [\text{V}] \tag{5}$$

The chosen operational amplifier (op-amp) (MAX44263AXA+T, Analog Devices Inc., Wilmington (MA), USA) was selected for its low bias current, offset voltage, power consumption, and input capacitance, along with a large gain-bandwidth product (GBP). Its rail-to-rail input/outputs maximize dynamic range for precise analog-to-digital converter (ADC) readouts. The op-amp comes in a dual SC70 package enabling a two-stage design to mitigate the impact of a single large feedback resistor $R_F$, which can introduce parasitic capacitance and increase noise contribution.

The first stage acts as a transimpedance amplifier with a gain of up to 5 MV A$^{-1}$, while the second stage functions as an inverting amplifier with a gain of 10 V V$^{-1}$, potentially resulting in a cumulative transimpedance gain of 50 MV A$^{-1}$. The schematic of the amplifier circuit is depicted in Fig 3.

The op-amp operates on a single 5V direct current (DC) power supply ($V_{supply}$) with two stabilizing capacitors $C_4 = 0.1$ µF and $C_5 = 1$ µF in parallel that filter noise, stabilize voltage fluctuations, and provide localized energy reservoirs to ensure stable and reliable operation of the TIA by minimizing ground bounce and preventing parasitic oscillations that could affect the output.

To establish a linear dynamic range and prevent saturation at the negative power supply in the absence of photocurrent, a bias of 2.7V is selected for the non-inverting inputs. This bias voltage serves a dual purpose: it facilitates an inverting amplifier stage crucial for single power supply operation where negative voltage swing cannot be achieved, and it establishes a virtual ground reference point, determining the maximum voltage swing of the TIA.

Single-supply op-amp with a midway bias voltage ensure efficient operation within their specified parameters. While a higher bias voltage might offer a larger inverting range and potential for greater amplification, it could also increase noise levels and cause earlier saturation for larger signals.

The power supply voltage and the bias voltage for photoconductive mode collectively determine the amplifier's effective operating range:

$$0V \leq V_{out} + V_{bias} \leq V_{supply}. \qquad [\text{V}] \qquad (6)$$

Achieving this operating range involves utilizing a voltage divider consisting of resistors $R_2$ = 22 kΩ and $R_3$ = 27 kΩ. To mitigate noise from the voltage divider and power supply, a capacitor $C_S$ is added in parallel with resistor $R_3$. With a value of 1 μF, this capacitor establishes a corner frequency $f_b$ (-3 dB) per Eq (7), eliminating frequency noise above 13 Hz:

$$f_b = \frac{1}{2\pi \cdot R_2 \| R_3 \cdot C_S}. \qquad [\text{Hz}] \qquad (7)$$

The electrical behavior of the photodiode under zero bias is represented by a current source and a junction capacitance $C_J$ of 60 pF as specified by the manufacturer. This inherent capacitance contributes to instability and noise in the amplifier. To counteract this effect and minimize high-frequency noise at the op-amp output, feedback capacitors $C_F$ are introduced. The initial feedback capacitor $C_{F1}$ compensates for both the photodiode's junction capacitance and the op-amp's input capacitance and is set to 6.8 pF. The feedback resistor $R_{F1}$, a potentiometer ranging up to 5 MΩ, is initially set at its maximum value, resulting in a bandwidth (-3 dB) of at least 4.68 kHz:

$$f_{-3\text{dB}} = \frac{1}{2\pi \cdot R_F \cdot C_F}. \qquad [\text{Hz}] \qquad (8)$$

With an estimated maximum photocurrent of 432 nA, the amplifier configuration provides ample headroom: to ensure that the cumulative transimpedance gain is sufficient even for small incident powers and map the obtained output voltage of the photodiode $V_{out}$ with a bias voltage of 2.7V, a total gain of 6 MV A$^{-1}$ is sufficient according to Eq (5). With a set gain of $G_2$ = 10 V V$^{-1}$ for the second amplifier stage, a gain of $G_1$ = 600 kV A$^{-1}$ ($R_{F1}$ = 600 kΩ) is needed for the first stage. As the value of $R_{F1}$ is reduced, the bandwidth of the circuit increases. Alternatively, the feedback capacitor can be exchanged for a larger capacitor.

Notably, a substantial portion of the gain is allocated to the initial transimpedance stage. The noise contribution from the second amplifier stage is relatively lower compared to the first stage. To minimize high-frequency noise in the second stage, another 6.8 pF capacitor $C_{F2}$ is introduced in parallel with resistor $R_{F2}$ to roll-off high-frequency gain. Since the cumulative transimpedance gain is the product of $R_{F1}$ and $R_{F2}$ divided by $R_4$, $R_{F2}$ is chosen as 470 kΩ, resulting in a bandwidth of 50 kHz according to Eq (8). Consequently, $R_4$ is set to 47 kΩ.

Due to the high gain of the amplifiers, an output coupling capacitor $C_6$ = 10 μF is required, placed in series with the output of the first op-amp. This capacitor blocks DC voltages and allows only alternating current (AC) signals to pass, ensuring that steady-state signals such as ambient light or constant offsets from electrical components are not amplified. Only the pulsating signals from the illuminated tissue are processed. This capacitor, in conjunction with the resistor, creates a time constant $\tau$ due to the RC behavior of the circuit:

$$\tau = R \cdot C. \qquad [\text{s}] \qquad (9)$$

The parameter $\tau$ corresponds to the time it takes for the capacitor to charge to approximately 63% of its maximum voltage. In our setup, a large time constant of $\tau$ = 0.47s results. This ensures an almost flat pulse response, as the slow charging of the capacitor smooths out rapid voltage changes. However, when the circuit is first turned on, the capacitor $C_6$ takes about two seconds to charge to the bias voltage and establish equilibrium (Fig 4(a)). During this initial

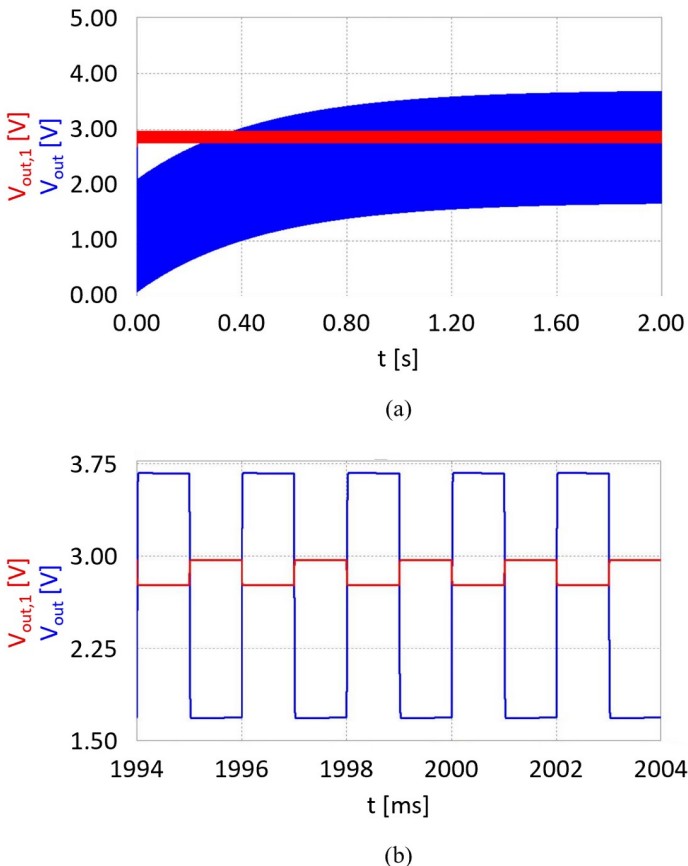

**Fig 4. Response simulation of the two-stage, low-noise transimpedance amplifier circuit with 2.7V bias for two alternating square wave signal inputs with amplitudes of 432 nA and 101 nA, a frequency of 500 Hz, and 50% duty cycle each.** The output voltage after the first stage ($G_1 = 600$ kV A$^{-1}$) is illustrated in red; the output voltage after the second stage ($G_2 = 10$ V V$^{-1}$) is illustrated in blue. (a) Simulation over the first two seconds showing the system's initial behavior. The settling time of the output capacitor $C_6$ is evident from the increasing output voltage over time, reaching equilibrium at the bias voltage. (b) Simulation showing the system's steady-state behavior. Peak-to-peak voltages are $\Delta V_{out,1} = 198.6$ mV and $\Delta V_{out} = 1.986$ V, fluctuating around the bias voltage of 2.7V.

charging period, false voltages may be outputted. This phenomenon is referred to as the capacitor's settling time or initialization phase [36].

The AC coupling introduced may affect both the frequency response and transient behavior of the circuit. In cases where the gain of the second-stage op-amp is low, this capacitor may be bypassed, given that the amplification of offset and noise is insignificant.

We simulated the transient response of the TIA in Micro-Cap 12 (Spectrum Software, Sunnyvale (CA), USA) to verify its operational principle. For the simulation, two alternating square wave signals with an amplitude of 432 nA (photodiode current estimated for 940 nm on cancellous bone) and 101 nA (photodiode current estimated for 1310 nm on cancellous bone), a frequency of 500 Hz, and 50% duty cycle each are generated to test the amplifier's time response. The simulated response for both op-amp stages is illustrated in Fig 4.

For a first amplifier stage gain of 600 kV A$^{-1}$, the difference in input signal $\Delta I_{PD}$ leads to a peak-to-peak voltage of

$$\Delta V_{out,1} = \Delta I_{PD} \cdot G_1 = 198.6 \, \text{mV}.$$

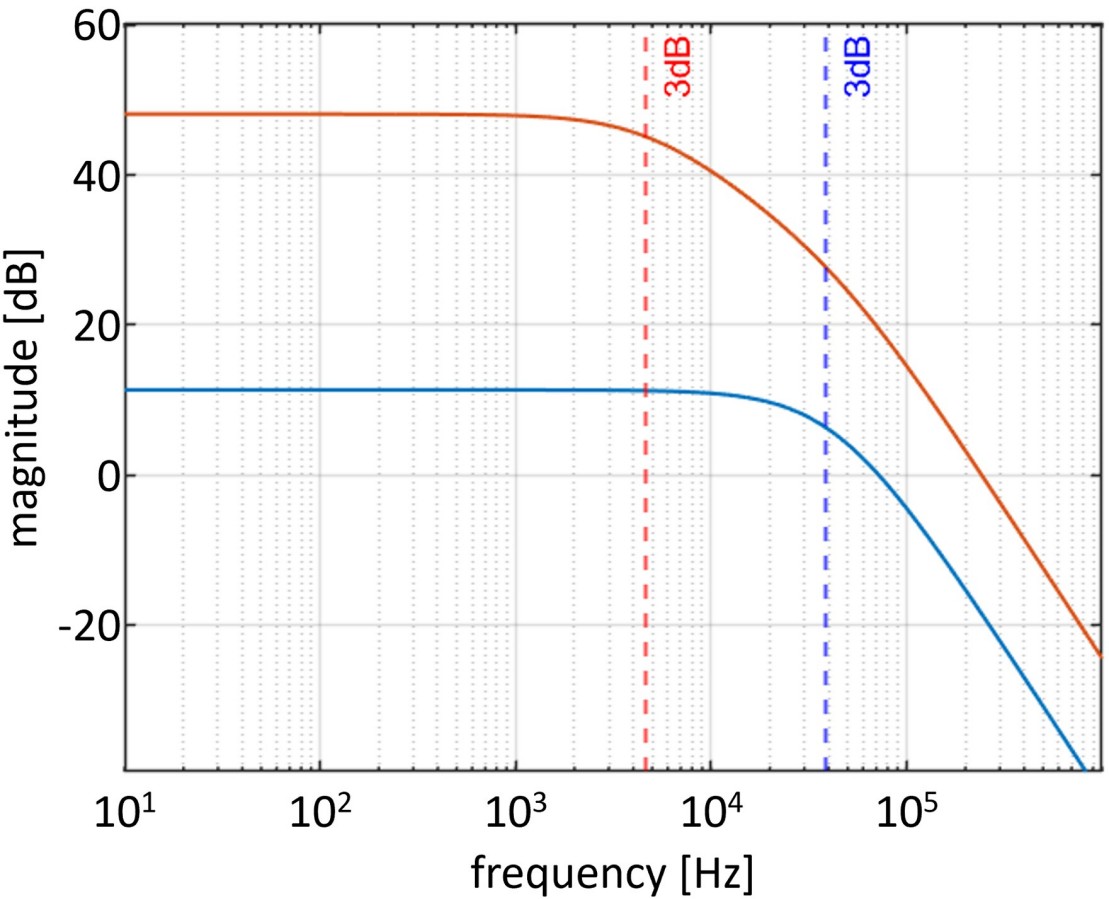

**Fig 5. Frequency response simulation of the two-stage, low-noise transimpedance amplifier circuit, indicating a bandwidth of 4.68 kHz for a total gain of 50 MV A⁻¹ (red) and a bandwidth of 39 kHz for a total gain of 6 MV A⁻¹ (blue).**

The second stage is an inverter and amplifies the signal towards ground, resulting in a peak-to-peak voltage of

$$\Delta V_{out} = \Delta V_{out,1} \cdot G_2 = 1.986 \text{ V}.$$

The output voltage is fluctuating around the bias voltage of 2.7V, yielding outputs of $V_{out} = 3.693$ V and $V_{out} = 1.707$ V as visualized in Fig 4(b).

The simulated frequency response of the amplifier is shown in Fig 5, where the amplitude of the test input signal is swept from 10 Hz to 1 MHz. For a total gain of 50 MV A⁻¹ ($G_1 = 5$ MV A⁻¹), the bandwidth is limited to 4.68 kHz by the first amplifier stage (red curve); for a total gain of 6 MV A⁻¹ ($G_1 = 600$ kV A⁻¹), the bandwidth is limited to 39 kHz by the first amplifier stage (blue curve).

## Electronics

The electronics comprise the driver circuits for the laser diodes and the signal amplification of the detected photodiode current, implemented on a double-sided PCB, Fig 1 ③ designed using KiCad 7.0 software, available at https://www.kicad.org/. The PCB is constructed from

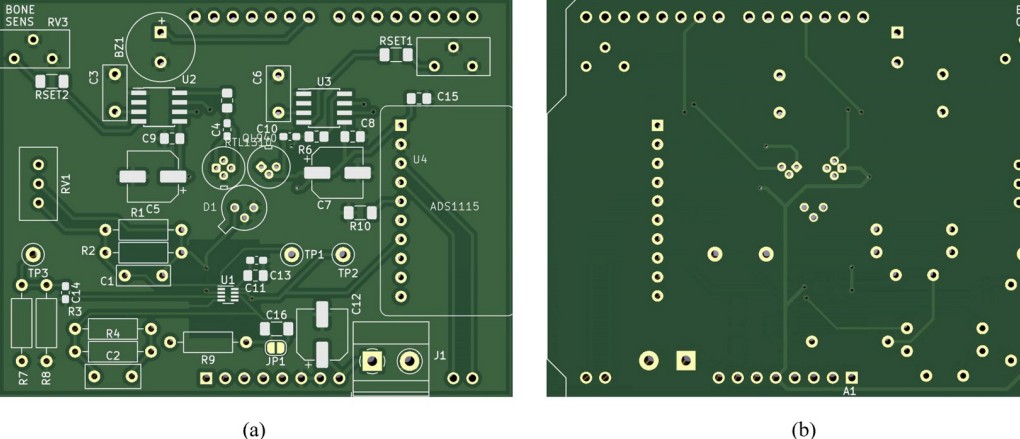

**Fig 6.** PCB layout: (a) Front view (b) Back view. Conductive pathways connecting nodes are visible within the ground pour. surface-mount device (SMD) pads are depicted in silver, and through-holes are represented as golden circles. Components outlines are highlighted in white. The corresponding schematic can be found here: doi.org/10.4121/77363d00-f68c-4da7-b32e-3aabe957ef98.

FR4 material with a thickness of 1.6 mm, and finished using the common and cost-effective hot air solder leveling (HASL) process with lead technology.

The PCB features varying tracing widths based on component requirements. Notably, the dual SC70 op-amp package necessitates a 0.3 mm tracing width due to its small pin spacing, while the 5V supply line employs a wider 0.5 mm width. All other traces are set at 0.4 mm.

The layout of the PCB prioritizes efficient design. Laser diodes and the photodiode are centrally positioned to facilitate optical fiber coupling without excessive bending. Nearby placement of related components minimizes potential tracing issues. To optimize op-amp performance, the ground pour around the op-amp area is removed, reducing parasitic capacitance to the inverting inputs.

To meet size constraints, most components are SMD, chosen for their compact packaging. However, for greater flexibility during prototyping, several components including feedback resistors, capacitors, bias voltage elements, the integrator capacitor, and potentiometers, are designated as through-hole rather than SMD, facilitating easy replacement if needed. A front and backside view of the PCB can be seen in Fig 6 with the corresponding PCB schematic found here: doi.org/10.4121/77363d00-f68c-4da7-b32e-3aabe957ef98.

The system can be powered via a USB supply from a connected laptop or similar device. Alternatively, a screw terminal labeled $J_1$, equipped with a bypass capacitor, allows connection to alternative power sources. Strategically placed test points ($TP_1$, $TP_2$, $TP_3$) facilitate measurements of bias voltage and amplifier output voltages.

## Control system

For this proof-of-concept, we selected an Arduino microcontroller board (Arduino Uno Rev3, Arduino SA, Chiasso, Switzerland), Fig 1 (4), for its extensive documentation, support, and ease of programming. The Arduino provides sufficient processing power and memory to handle all aspects of the device's intended functionality, including real-time control of laser diodes, readout and storage of photodiode output, computing for tissue classification, and operating the feedback system.

To convert the output voltage from the receiver's TIA into digital values, we use an external 16-bit ADC (ADS1115, Texas Instruments Inc., Dallas (TX), USA). This ADC offers higher resolution than the Arduino's internal 10-bit ADC, enabling finer granularity and increased accuracy in capturing the analog signals from the TIA. This enhanced resolution is crucial for accurately detecting the small analog signals from the photodiode. Additionally, the selected ADC features programmable gain settings, allowing amplification of weak signals without compromising resolution if needed. The ADC operates independently of the Arduino's processor, handling A/D conversion tasks and freeing up processing power for control functions.

The converted values are communicated to the Arduino using inter-integrated circuit (I2C) communication, which is facilitated by the SCL (serial clock) and SDA (serial data) inputs of the Arduino. The SCL line provides a clock signal to synchronize data transfer, while the SDA line facilitates bidirectional communication. In this setup, the Arduino acts as the master device, controlling the communication process by initiating data transmission, generating the clock signal, and addressing the ADC to send or receive information.

The ADC is mounted on the PCB. To establish a stable connection between PCB and Arduino, male headers are soldered onto the PCB, matching the pin configuration of the Arduino's female headers.

## Feedback system

In the operating room, efficient feedback representation is vital. While visual feedback offers comprehensive data presentation, audio feedback, with its capacity to encode one-dimensional data through pitch variation, proves particularly effective in capturing the surgeon's attention without diverting focus from the surgical site, as demonstrated by the PediGuard [37, 38].

We chose a constant tone buzzer (Multicomp Pro MP-ABI-050-RC, Premier Farnell Ltd., Farnell, UK), Fig 1 (5), for audio feedback with a sound output of 75 dB. The buzzer uses a piezoelectric element controlled by modulating the digital output to produce distinct auditory signals. The buzzer is mounted on the PCB.

## Casing & assembly

The diodes are securely housed within a custom 3D-printed diode holder, Fig 1 (6a), manufactured using a digital light processing (DLP) printer (Perfactory 4 Mini XL, EnvisionTEC GmbH, Gladbeck, Germany) and HTM 140 V2 resin. They are fixed in place by a lid (6b) fabricated from a stainless steel (AISI 301) sheet using wire electrical discharge machining (EDM). The diode holder and lid are connected by one M2x5 screw and two dowel pins. A stainless steel (AISI 304) capillary tube with an outer diameter of 2.1 mm and an inner diameter of 1.9 mm serves as the shaft (6c) that guides the optical fibers to the probe tip (6d). The tip, made from stainless steel (AISI 316), was crafted on a lathe machine, with holes for the fibers created using wire EDM.

All components are embedded in a two-part housing comprising a tip, Fig 1 (7a), and a handle (7b) resembling a screwdriver grip. The flat section of the tip part is milled from stainless steel (AISI 316) with a slot created using wire EDM. The pointed section of the tip part as well as the entire handle part are 3D-printed using DLP with HTM 140 V2 resin. The two tip sections are glued together, and both parts (tip and handle) are connected by three M2.5x20 screws. The Arduino is mounted to the handle using four M3x10 screws and nuts. Fig 7 shows the fully assembled probe.

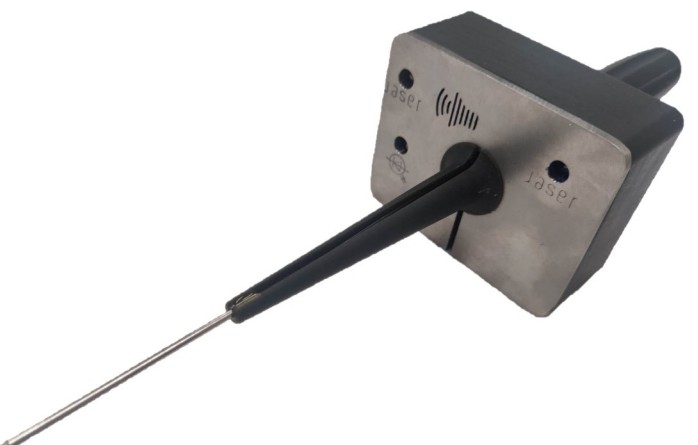

**Fig 7. Photo of the fully assembled handheld probe.**

## Operation of the system

At startup, the system initializes by configuring ports and communication protocols. During operation, the Arduino generates 5V pulse-width modulation (PWM) signals on pins 6 and 11 to control the laser diodes via their driver ICs. The PWM frequency is set to 55 Hz with a 50% duty cycle. This frequency is synchronized with the ADC readout (55 samples per second (SPS)) to ensure simultaneous activation and readout of each of the laser diodes.

Human auditory response times are typically around 200 ms [39, 40]. Consequently, the user output is set at 5 Hz, and the resulting digital values $x$ from the first (940 nm) and second (1310 nm) laser diode pulses are collected and averaged every 200 ms to calculate the reflection ratio $Q$:

$$Q = \frac{\bar{x}_{940}}{\bar{x}_{1310}}. \qquad [-] \tag{10}$$

The Arduino activates the buzzer connected to pin 8 based on the comparison between $Q$ and a set threshold ratio $Q_t$. Based on the expected photocurrents, $Q_t$ is set to 4.75. If the currently measured ratio exceeds the threshold ratio, the buzzer is activated, indicating contact with cortical bone; otherwise, it remains inactive.

## Validation

### Driver circuit

To assess whether the driver circuit is adjustable and able to autonomously regulate the operating current, the input pin IN on the IC mounted on the PCB was connected to a regulated 5V DC power supply for continuous operation. A spare NIR laser diode with a typical operating current of 20 mW and a monitor current of 0.2 mA was used for testing. The IC achieved a stable operating current of 20 mA, as expected, and adjustments to the $R_{SET}$ resistor demonstrated effective current regulation.

To validate proper functionality, an artificial monitor current of 0.5 mA (the maximum allowable for the IC) was generated using a 5V source and a 10 kΩ resistor, simulating high incident power on the photodiode integrated within the laser diode. Applying this current to pin AMD on the IC decreased the operating current and reduced the optical output power,

confirming the driver's ability to autonomously regulate current to ensure consistent light output for reliable feedback even over extended periods.

## Signal amplification

To validate the AC-coupled TIA, we generated a small AC signal simulating the expected photocurrent using a function generator with an 8.3 MΩ resistor in series, approximating a photocurrent of 337 nA with a bias voltage of 2.8V. A 300 Hz square waveform with a 50% duty cycle was applied to the input of the TIA (negative pin of the first-stage op-amp), and the amplified signal was observed using an oscilloscope (TDS2022B, Tektronix, Beaverton (OR), USA) connected to the output of the two-stage TIA. The system's response is shown in Fig 8 (a). The input frequency was then swept from 300 Hz to 1.5 kHz to observe changes in gain or waveform distortion, see Fig 8(b).

The gain of the first stage $G_1$ was set to 440 kV A$^{-1}$, and the second stage inverts and amplifies the signal by a factor of $G_2 = 10$ V V$^{-1}$. The measured output voltage was around 1.4V, indicating a slight deviation from the theoretical output voltage of 1.32V. This discrepancy was consistent among input signal frequencies and may be attributed to minor variations in photocurrent estimation or gain measurement accuracy.

For a photocurrent of 337 nA, the maximum output swing is achieved at 831 kV A$^{-1}$. Increasing the gain beyond this value resulted in output saturation, as shown in Fig 8(c). The AC coupling was verified by confirming that the TIA's output returned to bias when a DC signal was applied.

## Assembled prototype

An initial measurement for the assembled prototype was taken with both laser diodes turned off to establish a baseline and confirm a steady output. We expected $V_{out}$ to be 2.7V, corresponding to the selected bias voltage. The device showed steady ADC readings of $x_{940}$ and $x_{1310}$ around 2755, confirming correct bias voltage and no influence from ambient light.

To assess the butt-coupling efficiencies of the diodes, a power meter (S132C + PM100D, Thorlabs Inc., Newton (NJ), USA) measured the output power from the tip of the prototype for both laser diodes in a dark environment. The laser diodes were operated individually in constant mode by holding pin IN on the IC at 5V. The measured incident powers were 7.2 mW for the 51.1 mW 940 nm laser diode (coupling efficiency of 14%) and 80 μW for the 10.3 mW 1310 nm laser diode (essentially no light coupling).

A spare NIR laser diode with a nominal output power of 50 mW was placed in front of the tip of the prototype and operated at 55 Hz with a 50% duty cycle to assess photodiode coupling. No incoming signal was registered ($x = 2755$), indicating no light coupling for the photodiode.

To verify correct functioning of our assembled prototype, apart from the optical coupling, we replaced the 940 nm laser diode with a fiber-coupled laser diode of comparable properties (same wavelength and output power) including a single-mode fiber with 9 μm core diameter (SPL940-50-9-PD, Roithner Lasertechnik GmbH, Vienna, Austria). The photodiode is replaced by a fiber-receptacle photodiode with an active area of 1 mm diameter (FCPD-1000-FC, Roithner Lasertechnik GmbH, Vienna, Austria). The photodiode is connected to a 400 μm core diameter optical fiber with an NA of 0.22 through an FC/PC connector (M146L02, Thorlabs Inc., Newton, NJ, USA). The fibers are included in a modified probe tip that matches the new fiber diameters while maintaining the SDS at 1.3 mm.

The fiber-coupled laser diode was activated using a PWM frequency of 55 Hz with a 50% duty cycle. The probe was positioned 3 mm away from a Spectralon white reference standard

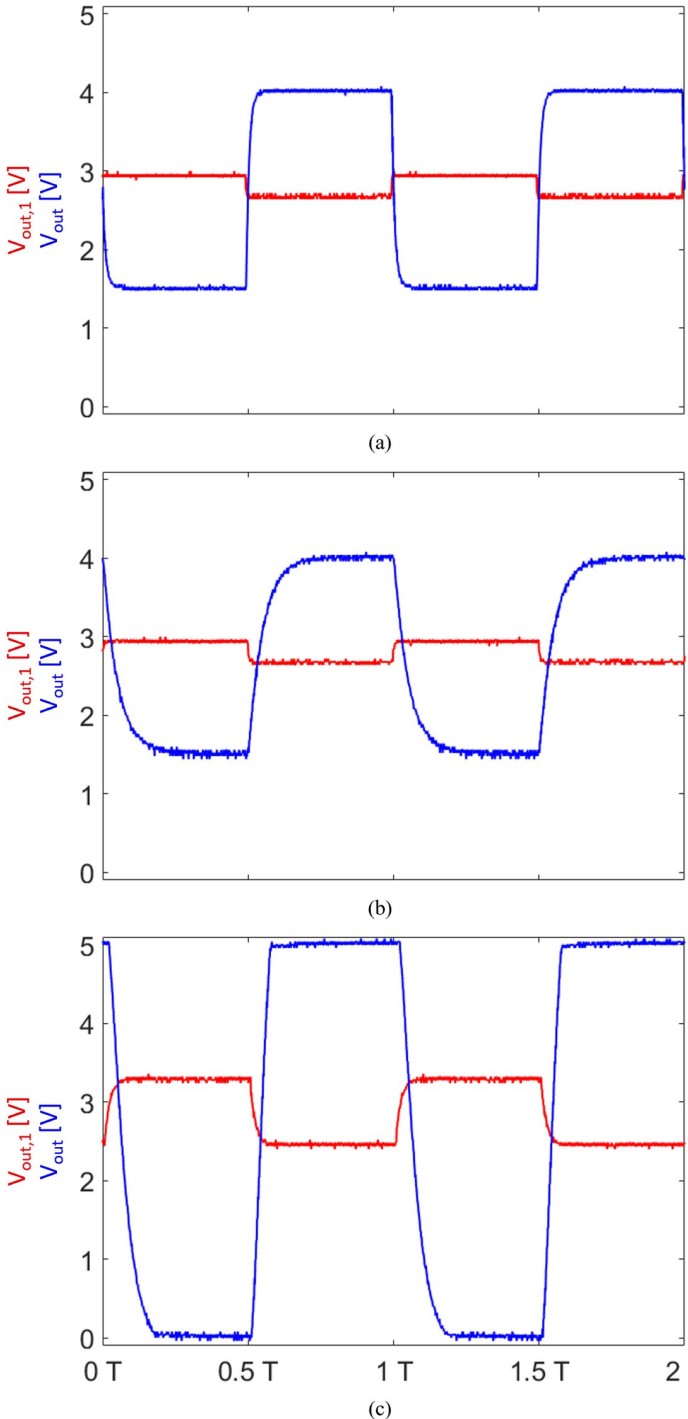

**Fig 8. Response of the two-stage, low-noise transimpedance amplifier circuit with 2.8V bias for a square wave signal input with an amplitude of 337 nA, recorded using an oscilloscope.** The output voltage after the first stage is illustrated in red; the output voltage after the second stage is illustrated in blue. (a) Input signal frequency of 300 Hz with a 50% duty cycle, $G_1 = 440$ kV A$^{-1}$. (b) Input signal frequency of 1.5 kHz with a 50% duty cycle, $G_1 = 440$ kV A$^{-1}$. (c) Input signal frequency of 1.5 kHz with a 50% duty cycle, $G_1 \geq 831$ kV A$^{-1}$.

(WS-1-SL, Labsphere Inc., North Sutton (NH), USA) to assess the device's optical functionality. With amplification set at its highest setting of 50 MV A$^{-1}$ and 99% reflection from the reference standard, we expected $V_{out}$ to drop and saturate at 0V. When the 940 nm laser was activated, the digitized reflectance value was indeed $x_{940} = 0$, confirming that photons are emitted by the 940 nm laser diode, reflected off the white reference standard, and collected by the photodiode.

Next, the probe was placed in cancellous bone-mimicking phantom (pure coconut milk with a fat content of 17% [23]), providing a highly controlled environment, allowing us to isolate and validate the technical performance of the device without the biological variability inherent in ex-vivo or in-vivo tissue. Reflectance was measured, and with an amplification of 50 MV A$^{-1}$, the digitized reflectance value was saturating at $x_{940} = 0$. Reducing the amplification to 6 MV A$^{-1}$ resulted in a digitized reflectance value of $x_{940} \approx 500$, demonstrating that even with low tissue reflectance, our device can collect a measurable signal.

## Discussion

In this paper, we presented a handheld fiber-optic tissue sensing device for spine surgery. Our design prioritizes size, cost, and speed, and comprises two laser diodes as the light source, a photodiode as the photodetector, a double-sided PCB for the electronic circuits, an Arduino Uno as the control unit, a buzzer for audio feedback, and a casing to house all components within a handheld device. The device generates real-time audio feedback to help surgeons prevent pedicle screw breaches and improve overall accuracy during spinal fusion.

The limited availability of compact and cost-effective laser diodes with the required wavelengths led us to use manually fiber-coupled laser diodes with butt-coupling in our prototype, resulting in low coupling efficiencies. Despite its simplicity and common use in practical device implementation, butt-coupling is the least efficient method for coupling a light source to a fiber because it does not match the fiber's numerical aperture. The high divergence of laser light emitted from the diode's active area causes a significant portion of light to fall outside the fiber core's diameter and acceptance angle, leading to leakage and only a fraction being permanently guided through the fiber [34, 41], and potential misalignments further reduce throughput efficiency. Additionally, excessive optical adhesive and reflections within the diode holder or housing can cause light to reach the sensor without interacting with the tissue. By replacing butt-coupled diodes with off-the-shelf fiber-coupled alternatives, our research demonstrated successful light emission and collection from tissue phantom within a handheld device. However, purchasing fiber-coupled laser diodes imposes constraints regarding size, wavelength, and fiber type/diameter, and can be significantly more expensive, especially for customized components.

To mitigate these shortcomings, reliable assembly tools and processes for butt-coupling have to be established to align the fiber and diode surfaces with high precision. Innovative methods can help with the alignment of the optical fiber and the diode. For example, integrating optical fibers during 3D printing through continuous fiber reinforcement (CFR) can be advantageous. The Mark Two printer (Markforged, Watertown (MA), USA) can reinforce 3D printed parts with carbon, fiberglass, or Kevlar fibers [42], which could facilitate the integration of fiber optics. Alternatively, tapered micro lenses offer better coupling efficiency by addressing both internal factors (such as laser wavelength and beam waist radius, lens shape and refractive index, fiber diameter and refractive index) and external factors (like alignment errors) [43].

While our prototype specifically focuses on spine surgery, DRS has been demonstrated as a useful tool in other body regions like the colon, lungs, or breast [44–47].

## Size considerations

The bone sensing device was designed so that the PCB functions as a shield for the Arduino Uno, with diodes positioned for easy connection and coupling of the electronics and optics. By optimizing the PCB layout and placing components on both sides, a smaller and more compact housing can be achieved, improving the handling of the handheld probe. Integrating a microcontroller onto the PCB and using only SMD components can further reduce the overall size.

The handheld device is currently powered via a USB supply, with the alternative option to power it via a screw terminal. For autonomous operation, a battery is required to power the device. The op-amp consumes a maximum of 2 mA. The 940 nm laser diode operates at 60 mA, and the 1310 nm laser diode operates at 35 mA. Since the laser diodes are pulsed at a 50% duty cycle, their effective current draw is halved. The ADC uses 150 μA in continuous mode, and the IC consumes up to 15 mA. This results in a total current consumption of approximately 65 mA for the optics and electronics. The Arduino Uno has an idle current consumption of 50 mA, increasing to an estimated maximum of 100 mA when running code. Consequently, the total current consumption for the device is approximately 165 mA.

Spinal fusion surgery can take up to 8 hours, depending on the extent of damage, the method of fusion, and the surgical approach used [48, 49]. To ensure a safety margin, the battery should last for at least 12 hours. Therefore, a battery capacity of 1980 mA h at 5V is required. This can be provided by a suitable battery pack (*e.g.*, 5V 18650 Lithium Ion Battery Pack 2200mAh, Himax Electronics Co. Ltd., Shenzhen, China), a 1.5V AA battery with sufficient capacity and a boost converter (*e.g.*, U1V10F5, Pololu, Las Vegas (NV), USA), or three AA or AAA batteries in series to achieve the necessary voltage. With the current design, the handle of the housing still leaves some space for a battery to fit in.

## Cost considerations

The optical and electronic components integrated into our handheld device are low-cost and readily available. By determining the output power requirements of the light source, we can explore the feasibility of replacing laser diodes with much cheaper LED to optimize costs without compromising device functionality. Although integrating optical components inherently increases production costs compared to electric-only devices like the PediGuard, this increase may be justified by the potential for superior performance. Additionally, alternative microcontroller options could offer cost-effective substitutes for the Arduino used in this proof-of-concept, further enhancing the device's affordability.

Currently, manufacturing and assembly efforts are significant factors in the overall cost. While initial prototype costs are higher than those in serial production, there is potential to reduce these costs. Soldering small components onto the PCB is currently done manually under a microscope, but with known component sizes, PCB assembly can be outsourced. Manufacturing EDM parts requires special equipment and training, so the design should be revised to employ cheaper manufacturing techniques such as computerized numerical control (CNC) machining or laser cutting. Although 3D printing is beneficial for prototyping or personalization of medical instruments [50], it is relatively slow and expensive for serial production. Therefore, printed parts should be redesigned for production-friendly manufacturing methods like injection molding.

## Speed considerations

The device's operational frequency, capped at 55 Hz by the Arduino's capabilities, presents a significant constraint. While the Arduino Uno is ideal for prototyping thanks to its user-

friendly interface and ease of programming, its 10-bit ADC resolution is insufficient for high-accuracy measurements, limiting its application in specialized designs. To mitigate this constraint, an external high-resolution ADC with a maximum sampling rate of 860 SPS was integrated. However, the slow communication between the external ADC and Arduino diminishes the device's operating speed. This communication bottleneck is compounded by I2C protocol delays, ADC conversion times, and software overhead, which collectively reduce the practical sampling rate.

To overcome these challenges, upgrading to a more powerful microcontroller like ARM (ARM Holdings Limited, Cambridge, UK) or ESP (Espressif Systems, Shanghai, China) variants offers superior processing power, memory, and versatile Input/Output (I/O) capabilities. Choosing the right microcontroller depends on specific application requirements for functionality, operational conditions, and packaging. Both options represent significant advancements over the Arduino Uno, ensuring a more robust and efficient system [51], next to cutting size and cost.

The current speed limitation results in a slow pulse repetition frequency. Increasing this frequency for faster probing allows to detect changes in signal more rapidly, and with higher pulse and sampling rates, larger datasets can be generated within the same time frame to facilitate smoother and more stable averaged signals for calculating the reflection ratio.

## Future work

We acknowledge that the current experiments are foundational and relatively simple, thus unable to fully assess the performance of our device in a clinical setting. However, as this work represents an early-stage proof-of-concept, the primary goal was to demonstrate the feasibility of this novel approach and validate our design through initial testing. Future optimization will involve more comprehensive evaluations, including ex-vivo trials, validation against conventional guidance technologies, and design iterations to fully validate its potential for clinical translation.

In the next step, the device should be validated according to its intended use by testing the prototype on cancellous and cortical bone samples from ex-vivo human or porcine vertebrae to replicate tissue sensing during spine surgical interventions. A sufficiently large sample of reflectance values ($\approx$ 50) should be obtained for both tissues, and their distributions should be assessed. The reflection ratio $Q$ for both cancellous and cortical bone should be computed for each reading and analyzed to verify the set threshold ratio and adjust it if necessary. The final threshold will be determined based on the results from experiments using human tissues. If $Q_t$ successfully separates the distributions of $Q_{canc}$ and $Q_{cort}$, it indicates that the two tissue types can be distinguished. Otherwise, a more complex metric than the reflection ratio may be necessary.

Furthermore, our study raises several considerations for the design and implementation of a handheld fiber-optic tissue sensing device. Variations in optical coupling efficiency among prototypes may influence the recorded signal, necessitating time-consuming calibration of each device to establish the threshold ratio. Ensuring uniformity in threshold ratio values across prototypes and procedures is crucial to avoid the need for recurrent recalibration by manufacturers or surgeons.

Additionally, in the future, our device will need to be validated against conventional guidance technologies in spine surgery, such as fluoroscopy and computer-assisted navigation. Notably, DRS has previously been benchmarked against Magnetic Resonance Imaging (MRI) for quantifying vertebral bone fat fraction [52], underscoring its potential to provide reliable tissue feedback. Previous work on ex-vivo human tissue and in-vivo swine models has established DRS as a viable tissue sensing technology for spine surgery [16, 20]. Comparative studies

will further establish the device's performance and its potential for integration into clinical practice.

Moreover, future iterations of our device must prioritize enhanced mechanical robustness to accommodate in-vivo operations, where surgeons apply considerable forces to the bone [53]. Drawing from experience in earlier ex-vivo and in-vivo tests of DRS probes [18, 20], we focused on securely coupling the optical fibers in this design. Redesign efforts should ensure that the device withstands the substantial mechanical stresses exerted on the handle and probe shaft during surgical procedures. In addition to mechanical robustness and sensing capabilities, usability in clinical practice is essential, and close collaboration with surgeons is encouraged to ensure the device meets clinical needs and integrates seamlessly into surgical workflows.

Design considerations extend to choosing between a single-use or a sterilizable device. While single-use devices mitigate contamination risks, they raise concerns about environmental sustainability. Achieving complete sterilizability poses challenges in ensuring the functionality of all components post-sterilization [54]. However, technologies for sterilizing fiber-based systems, such as fiber-optic endoscopes, are well-established [55], demonstrating that sterilization of optical components is feasible. Alternatively, a semi-reusable approach is imaginable, where only the shaft that comes in touch with the patient is disposable while the remaining components are reusable, balancing cost-effectiveness with environmental impact.

To further advance fiber-optic tissue sensing, several research directions could be explored. One potential improvement is multi-directional sensing, which would enable directional tissue feedback to detect breaches impending from various angles [56]. Another promising avenue is the integration of machine learning algorithms for real-time classification of tissue types, potentially increasing the speed and reliability of intraoperative feedback [57, 58]. Additionally, combining DRS with complementary sensing modalities, such as ultrasound or electrical impedance, could provide a more comprehensive assessment of tissue properties, broadening the applicability of fiber-optic sensing across various surgical specialties.

Eventually, implementing optical systems and fiber optic probes in medical environments requires regulatory approval and large-scale manufacturing of sterile probes, necessitating cost and complexity reduction. Future steps for translation into clinical settings include comprehensive clinical studies to validate device performance and usability for regulatory compliance and eventual commercialization. Biocompatibility of device components, particularly those in direct patient contact, must be guaranteed. Medical-grade stainless steel for the tip and biocompatible fibers [59] ensure patient safety and regulatory approval.

## Conclusion

Our handheld fiber-optic tissue sensing device utilizes DRS to distinguish bone tissues, demonstrating potential to provide surgical guidance in spinal fusion procedures. Thanks to its compact size and low cost, the device is well-suited for healthcare facilities with limited access to expensive surgical technologies, while also being compatible as an add-on to existing computer-assisted navigation or robotic-assisted systems. Despite initial challenges with fiber coupling, the device's compact design, affordability, and real-time feedback capabilities highlight its value as a potential tool in clinical settings, aimed at enhancing patient outcomes through improved pedicle screw placement accuracy.

## Acknowledgments

The authors express their sincere gratitude for the invaluable support provided by Paul Keijzer, Lars Leenheer, Remi van Starkenburg, and David Jager at DEMO of Delft University of

Technology. Without your practical advice and steady hands, this prototype would have never seen the light of day.

## Author Contributions

**Conceptualization:** Merle S. Losch.

**Data curation:** Merle S. Losch.

**Funding acquisition:** Jenny Dankelman, Benno H. W. Hendriks.

**Investigation:** Merle S. Losch, Benjamin E. Visser.

**Methodology:** Merle S. Losch, Benjamin E. Visser.

**Software:** Benjamin E. Visser.

**Supervision:** Jenny Dankelman, Benno H. W. Hendriks.

**Validation:** Merle S. Losch, Benjamin E. Visser.

**Visualization:** Merle S. Losch.

**Writing – original draft:** Merle S. Losch.

**Writing – review & editing:** Jenny Dankelman, Benno H. W. Hendriks.

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
