## [Decision Letter · Decision Letter 0]

15 Sep 2024

PONE-D-24-30394A handheld fiber-optic tissue sensing device for spine surgeryPLOS ONE

Dear Dr. Losch,

Thank you for submitting your manuscript to PLOS ONE. After careful consideration, we feel that it has merit but does not fully meet PLOS ONE’s publication criteria as it currently stands. Therefore, we invite you to submit a revised version of the manuscript that addresses the points raised during the review process.

**ACADEMIC EDITOR**: Reviewers' comments provide more detailed feedback. So, we invite major revisions to the submitted article.

We look forward to receiving your revised manuscript.

Kind regards,

M. Jagabar Sathik

Academic Editor

PLOS ONE

Journal Requirements:

"This work was supported by the grant: NWO-TTW 17553."         

“The author affiliated with Philips Medical Systems (B.H.W.H.) has financial interests in the subject matter, materials, and equipment, in the sense that he is an employee of Philips. None of the other authors have any financial relationship or conflict of interests.”

5. The author affiliated with Philips Medical Systems (B.H.W.H.) has financial interests in the subject matter, materials, and equipment, in the sense that he is an employee of Philips. None of the other authors have any financial relationship or conflict of interests.

6. Please note that your Data Availability Statement is currently can’t access the DOI direct link. If your manuscript is accepted for publication, you will be asked to provide these details on a very short timeline. We therefore suggest that you provide this information now, though we will not hold up the peer review process if you are unable.

Additional Editor Comments:

The presented article has good archival value; however, significant improvement is required in the experimental section and presentation. Reviewers' comments provide more detailed feedback. So, I invite major revisions to the submitted article.

Reviewers' comments:

Reviewer's Responses to Questions

**Comments to the Author**

1. Is the manuscript technically sound, and do the data support the conclusions?

Reviewer #1: Yes

Reviewer #2: Partly

Reviewer #3: No

2. Has the statistical analysis been performed appropriately and rigorously? 

Reviewer #1: Yes

Reviewer #2: No

Reviewer #3: No

3. Have the authors made all data underlying the findings in their manuscript fully available?

Reviewer #1: Yes

Reviewer #2: Yes

Reviewer #3: Yes

4. Is the manuscript presented in an intelligible fashion and written in standard English?

Reviewer #1: Yes

Reviewer #2: Yes

Reviewer #3: Yes

5. Review Comments to the Author

Reviewer #1: Dear Authors,

The manuscript "A handheld fiber-optic tissue sensing device for spine surgery" presents a novel device designed to assist in real-time differentiation of bone tissue during spinal fusion surgery. The device uses DRS to aid surgeons in accurately placing pedicle screws, which is crucial for the success of spinal fusion procedures. The handheld prototype employs laser diodes at two wavelengths (940 nm and 1310 nm) to illuminate the tissue, with a photodiode detecting the reflected light, providing real-time feedback through audio signals.

The design emphasizes compactness, cost-effectiveness, and real-time feedback in surgical settings. Despite challenges related to fiber-optic coupling, the proof-of-concept demonstrated successful differentiation between cancellous and cortical bone tissues. The device shows potential to improve surgical accuracy and patient outcomes, though further testing and validation, including mechanical stress tests and clinical trials, are recommended.

Below are my comments:

The manuscript describes using a handheld device for real-time bone tissue differentiation. Could the authors provide more details on the selection criteria for the wavelengths used (940 nm and 1310 nm)? Why were these wavelengths prioritized, particularly considering the broader NIR spectrum?

The paper discusses challenges related to optical coupling efficiencies. Could the authors elaborate on how the device was calibrated across different tissue types? Was any baseline established for cancellous and cortical bones prior to clinical testing?

The device aims to improve pedicle screw placement accuracy during spinal fusion. How was it ensured that the device's real-time feedback remains reliable and consistent over extended periods?

The design prioritizes cost and size, but surgical instruments typically experience significant mechanical stress. Have the authors conducted any stress tests or fatigue analyses to ensure the device’s functionality during extended or repeated use?

While the manuscript mentions plans for future clinical validation, it would benefit from discussing any preliminary testing or feedback from surgeons. Have any initial trials been conducted, and if so, what were the key insights or takeaways?

The manuscript references similar devices like PediGuard, which uses EIS. Can the authors elaborate on the advantages of DRS over EIS? In what specific clinical scenarios would DRS outperform EIS regarding surgical guidance?

Since this device is intended for surgical use, could the authors expand on the sterilization process for the components? How will the integrity of the optical fibers and electronics be maintained after repeated sterilizations?

Reviewer #2: The manuscript entitled “A handheld fiber-optic tissue sensing device for spine surgery” has an interesting subject and useful results and it can be published after some revision according to the below comments:

• Please improve the English language and structure

• Article paragraphing should be improved

• Several explanatory schematics corresponding to the involved mechanisms should be added

• What are the key advantages of using a handheld fiber-optic tissue sensing device compared to traditional methods for pedicle screw placement in spine surgery?

• How does Diffuse Reflectance Spectroscopy (DRS) contribute to the differentiation of bone tissues during surgical procedures?

• What specific challenges were encountered in the development of the handheld fiber-optic device, particularly regarding coupling efficiencies?

• How does the integration of laser diodes at two distinct wavelengths enhance the functionality of the tissue-sensing device?

• What role does the printed circuit board (PCB) play in the operation of the handheld fiber-optic tissue sensing device?

• How does the microcontroller process the signals from the photodiode to provide real-time feedback to surgeons?

• What are the implications of using low-cost and readily available components in the design of the fiber-optic sensing device for widespread clinical adoption?

• In what ways does real-time audio feedback improve surgical accuracy and outcomes during spinal fusion procedures?

• What are the limitations of the prototype device, and how might future iterations address these issues?

• How does the device's compact design influence its usability in various healthcare settings?

• What potential improvements could be made to the signal amplification process to enhance the accuracy of tissue differentiation?

• How does the device ensure that the feedback provided to surgeons is both timely and reliable during surgery?

• The literature should be improved using the below papers

o Wang, Y., Zhai, W., Zhang, H., Cheng, S., & Li, J. (2023). Injectable Polyzwitterionic Lubricant for Complete Prevention of Cardiac Adhesion. Macromolecular Bioscience, 2200554. doi: https://doi.org/10.1002/mabi.202200554

o Yao, R., Ge, Z., Wang, D., Shang, N., & Shi, J. (2024). Self-sensing joints for in-situ structural health monitoring of composite pipes: A piezoresistive behavior-based method. Engineering Structures, 308, 118049. doi: https://doi.org/10.1016/j.engstruct.2024.118049

o Fu, Y., Chen, X., Song, W., Kuang, J., Wu, W., Yang, X.,... Liao, Y. (2024). Light-Switch Electrochemiluminescence-Driven microfluidic sensor for rapid and sensitive detection of Mpox virus. Chemical Engineering Journal, 498, 154930. doi: https://doi.org/10.1016/j.cej.2024.154930

o Zhou, Y., Xie, J., Zhang, X., Wu, W., & Kwong, S. (2024). Energy-Efficient and Interpretable Multisensor Human Activity Recognition via Deep Fused Lasso Net. IEEE Transactions on Emerging Topics in Computational Intelligence, 1-13. doi: 10.1109/TETCI.2024.3430008

o Jiang, Z., Han, X., Zhao, C., Wang, S., & Tang, X. (2022). Recent Advance in Biological Responsive Nanomaterials for Biosensing and Molecular Imaging Application. International Journal of Molecular Sciences , 23(3), 1923. doi: https://doi.org/10.3390/ijms23031923

• What are the broader implications of integrating tissue sensing technology into surgical instruments for other types of surgeries beyond spinal procedures?

• How does the study address the need for improved surgical accuracy in the context of rising demand for interventions related to age-related degenerative diseases?

• What future research directions could be pursued to further develop and optimize fiber-optic sensing technologies in surgical applications?

Reviewer #3: This manuscript presents an interesting prototype—a handheld fiber-optic sensor for differentiating between bone and tissue. However, the authors did not compare the performance of the proposed prototype with conventional fiber-optic sensors. Additionally, the authors did not discuss the pros and cons of this technology compared to other modalities, such as ultrasonography, which is also portable and provides real-time imaging. The experiments are too simple, and the data is not robust enough to demonstrate the novelty of this work. Furthermore, Figures 2 and 3 are missing. The proposed concept is clinically useful. The reviewer suggests resubmitting the manuscript after adding more experiments (at least ex vivo).

6. PLOS authors have the option to publish the peer review history of their article (what does this mean?). If published, this will include your full peer review and any attached files.

Reviewer #1: No

Reviewer #2: No

Reviewer #3: No

---

## [Author Response · Author response to Decision Letter 0]

24 Oct 2024

Dear editor and reviewers,

Thank you for reviewing our manuscript and providing us with valuable comments. We have revised the article to address all comments, and the suggested alterations are incorporated in the revised manuscript (highlighted in blue). Please find our rebuttal in the attached response to the reviewers.

---

## [Decision Letter · Decision Letter 1]

15 Nov 2024

A handheld fiber-optic tissue sensing device for spine surgery

PONE-D-24-30394R1

Dear Dr. Losch,

We’re pleased to inform you that your manuscript has been judged scientifically suitable for publication and will be formally accepted for publication once it meets all outstanding technical requirements.

Kind regards,

M. Jagabar Sathik

Academic Editor

PLOS ONE

Additional Editor Comments (optional):

The authors incorporated all reviewer comments and suggestions. No further review.

Reviewers' comments:

Reviewer's Responses to Questions

**Comments to the Author**

1. If the authors have adequately addressed your comments raised in a previous round of review and you feel that this manuscript is now acceptable for publication, you may indicate that here to bypass the “Comments to the Author” section, enter your conflict of interest statement in the “Confidential to Editor” section, and submit your "Accept" recommendation.

Reviewer #2: All comments have been addressed

Reviewer #3: All comments have been addressed

2. Is the manuscript technically sound, and do the data support the conclusions?

Reviewer #2: Yes

Reviewer #3: Yes

3. Has the statistical analysis been performed appropriately and rigorously? 

Reviewer #2: Yes

Reviewer #3: Yes

4. Have the authors made all data underlying the findings in their manuscript fully available?

Reviewer #2: Yes

Reviewer #3: Yes

5. Is the manuscript presented in an intelligible fashion and written in standard English?

Reviewer #2: Yes

Reviewer #3: Yes

6. Review Comments to the Author

Reviewer #2: (No Response)

Reviewer #3: The authors have addressed the comments well. The revised manuscript is in a good shape to be considered for publication.

7. PLOS authors have the option to publish the peer review history of their article (what does this mean?). If published, this will include your full peer review and any attached files.

Reviewer #2: **Yes: **Shokouh Attarilar

Reviewer #3: No

---

## [Editor Report · Acceptance letter]

21 Nov 2024

PONE-D-24-30394R1 

PLOS ONE

Dear Dr. Losch, 

I'm pleased to inform you that your manuscript has been deemed suitable for publication in PLOS ONE. Congratulations! Your manuscript is now being handed over to our production team.

Kind regards, 

on behalf of

Dr. M. Jagabar Sathik 

Academic Editor

PLOS ONE